# Novel Nanocombinations of l-Tryptophan and l-Cysteine:  Preparation, Characterization, and Their Applications for Antimicrobial and Anticancer Activities

**DOI:** 10.3390/pharmaceutics13101595

**Published:** 2021-10-01

**Authors:** Ahmed I. Abd-Elhamid, Hamada El-Gendi, Abdallah E. Abdallah, Esmail M. El-Fakharany

**Affiliations:** 1Composites and Nanostructured Materials Research Department, Advanced Technology and New Materials Research Institute, City of Scientific Research and Technological Applications (SRTA City), New Borg El-Arab 21934, Egypt; ahm_ch_ibr@yahoo.com; 2Bioprocess Development Department, Genetic Engineering and Biotechnology Research Institute, City of Scientific Research and Technological Applications (SRTA City), New Borg El-Arab 21934, Egypt; 3Pharmaceutical Medicinal Chemistry & Drug Design Department, Faculty of Pharmacy (Boys), Al-Azhar University, Cairo 11884, Egypt; abdulla_emara@azhar.edu.eg; 4Protein Research Department, Genetic Engineering and Biotechnology Research Institute GEBRI, City of Scientific Research and Technological Applications (SRTA City), New Borg El-Arab 21934, Egypt

**Keywords:** modified nanocombinations, l-tryptophan and l-cysteine, antimicrobial, anticancer

## Abstract

Tungsten oxide WO_3_ nanoparticles (NPs) were prepared in a form of nanosheets with homogeneous size and dimensions in one step through acid precipitation using a cation exchange column. The resulting WO_3_ nanosheet surface was decorated with one of the two amino acids (AAs) l-tryptophan (Trp) or l-cysteine (Cys) and evaluated for their dye removal, antimicrobial, and antitumor activities. A noticeable improvement in the biological activity of WO_3_ NPs was detected upon amino acid modification compared to the original WO_3_. The prepared WO_3_-Trp and WO_3_-Cys exhibited strong dye removal activity toward methylene blue and safranin dyes with complete dye removal (100%) after 6 h. WO_3_-Cys and WO_3_-Trp NPs revealed higher broad-spectrum antibacterial activity toward both Gram-negative and Gram-positive bacteria, with strong antifungal activity toward *Candida albicans*. Anticancer results of the modified WO_3_-Cys and WO_3_-Trp NPs against various kinds of cancer cells, including MCF-7, Caco-2, and HepG-2 cells, indicate that they have a potent effect in a dose-dependent manner with high selectivity to cancer cells and safety against normal cells. The expression levels of E2F2 and Bcl-2 genes were found to be suppressed after treatment with both WO_3_-Cys and WO_3_-Trp NPs more than 5-FU-treated cells. While expression level of the p53 gene in all tested cells was up-regulated after treatment 5–8 folds more as compared to untreated cells. The docking results confirmed the ability of both NPs to bind to the p53 gene with relevant potency in binding to other tested gens and participation of cysteine SH-functional group in such interaction.

## 1. Introduction

Tungsten oxide (WO_3_) is considered as one of the most promising transition metal oxides, for their performance in various applications, including electrochromic devices [1,2], dye-sensitized solar cells [3,4], photocatalytic applications [5], sensing applications [6,7], field-emission applications [8], high-temperature superconductors [9], optical recording devices [10], and adsorbent [11]. The wide application for WO_3_ could be attributed to its high chemical stability and remarkable electric conductivity in addition to the ability in the reverse-redox process [12]. Moreover, tungsten oxide could be prepared in different morphological structures, including nanorods [13], nanotubes [14], nanosheets [7], nanowires [15], and nanobelts [16]. Several techniques were investigated for the preparation of WO_3_ using either physical or chemical approaches; thermal evaporation [17], spray pyrolysis [18], sol–gels [19], the templating method [20], hydrothermal [13], electrochemical anodization [21], electrodeposition [12], and the microwave-assisted method [5].However, these methods are time-consuming, and relatively complicated processes with a limitation in controlling the size and shape of the resulted nanoparticles [5,22]. One of the simplest and cost-effective methods for preparing tungsten trioxide nanoparticles is acid precipitation [23]. Great attention has recently been directed toward increasing the nanometals implementation in various industrial sectors attributed to its tiny size and large surface-to-mass ratio, besides its remarkable stability in many harsh conditions [24,25]. Nonetheless, the high biotoxicity of some well-known NPs (Ag, Cu, Ti, etc.) has restricted their wide commercial usage [26,27]. To overcome the toxicity issue, attention was directed toward NP-oxides as a safer and effective alternative [28]. In the same regard, NPs surface modification was also proposed to increase the NPs applicability range with lower side toxicity [29,30]. Recently, growing interest in nanoparticles surface functionalization with different functional groups and bio-macules as amino acids, short peptides, and proteins toward new applications has been intensively reported [29,31]. Owing to the large surface-to-mass ratio, NPs adsorb these biomolecules on their large surface area in corona-like structure through Van der Waals forces, revealing new biological activities [32].

Nowadays, surface-functionalized NPs are efficiently applied in many fields, including environmental and medical fields. With growing dye-dependent industries, synthetic dye effluents represent great environmental and public health challenges attributed to their significant toxicity [25,33,34,35]. Though many approaches were proposed for dye remediation [33], the lower efficiency and high cost involved limiting their commercial applications [36]. Biotechnological applications of various NPs in hazardous chemicals remediation is an emerging field [25], with advantages of surface modification that enhance their dye-removal capacity [37]. Cancer has recently been considered as a global health threat representing the world’s second leading cause of death, with the expectation of becoming the leading cause of death in future years [38,39]. Cancer treatment, through the currently available protocols, usually results in immunosuppression for treated patients, a condition that renders them at high risk for severe microbial infections [40]. Developing a new anticancer agent with consolidated antimicrobial activity is a pressing need. To this end, inorganic nanometals represent one step forward, with their high antitumor and antimicrobial activity with minor bacterial-resistance evolution ability [41]. Moreover, the high stability of NPs toward harsh conditions, including temperature and pH intensifies the efforts for its successful medical applications [42]. Short peptides and amino acids have also attracted attention as promising antimicrobial agents [43]. Currently, there is wide interest in individual amino acid (AA) interaction with different NPs for better understanding and elucidation of corona-like structures and to give a deeper insight into the new resulting biological activities [44,45]. Cysteine and tryptophan-rich short peptides have revealed potent antimicrobial activity against many human pathogens [46,47]. The current study concerns the preparation and functionalization of WO_3_ NPs with tryptophan or cysteine amino acids and evaluates the efficacy of the resulting NPs-AAs in versatile applications, including synthetic dye removal, antimicrobial, and antitumor activities.

## 2. Material and Methods

### 2.1. Preparation of WO_3_ NPs

The WO_3_ NPs were prepared by passing 10 mL (0.5 M) of Na_2_WO_4_ (>98%, Sigma–Aldrich, Taufkirchen, Germany) solution through Ion exchanger Dowex^®^ 50 WX 4 (strongly acidic cation exchanger), H^+^ form, (obtained from Merck, Darmstadt, Germany) packed column. The yielded yellow solution was collected from the column in a blue-capped bottle at a flow rate of 10 mL/min using phosphate buffer saline (PBS, pH 7.2). Thereafter, the bottle was placed in an oven at 50 °C for 24 h (Figure 1A). The produced precipitate was separated by filtration, washed several times with double distilled water (ddH_2_O), dried at 70 °C, and finally, stored in a dark bottle for further use.

### 2.2. Surface Functionalization of WO_3_ NPs with Tryptophan or Cysteine

Surface functionalization of the prepared WO_3_ was conducted through mixing equal volumes of the prepared WO_3_ NPs (0.2 mg/mL) with tryptophan or cysteine (0.05 mg/mL) in 0.1 M phosphate pH 7.0 with stirring for 6 h. The mixtures were then centrifuged to eliminate the unbound amino acids, and washed several times with ddH_2_O and PBS, pH 7.2. Finally, the resulting NP-AA pellets were used during the following experiments (Figure 1B).

### 2.3. Characterizations of WO_3_ NPs and Surface-Functionalized WO_3_-AAs

The pristine WO_3_ and surface functionalized WO_3_-AAs were characterized using scanning electron microscopy (SEM; JSM 6360LA, Tokyo, Japan), Fourier transform-infrared (FT-IR) analysis using Shimadzu FTIR (Model, FTIR8400, Kyoto, Japan), thermal gravimetric analysis (TGA; Shimadzu Thermal Gravimetric Analysis, Kyoto, Japan), and the elemental EDS unit related to SEM. 

### 2.4. Application of WO_3_ NPs and Functionalized WO_3_-AAs in Synthetic Dyes Removal

The ability of the modified WO_3_-AAs in synthetic dyes remediation was evaluated against some synthetic dye solutions and compared to plain WO_3_ NPs and separate amino acids. Dye solutions (100 ppm/final conc.) were prepared by dissolving methylene blue and safranin in 0.1 M phosphate buffer, pH 7.0. Dye decolorization test was performed at a final volume of 5.0 mL by separately adding 0.4 mg/mL of NPs, WO_3_-AAs, or AAs to each dye solution. The reaction mixtures were incubated at room temperature for 6 h. Samples were withdrawn from each tube every 1 h and measured using a UV-vis spectrophotometer at wavelengths of 662 nm and 495 nm for methylene blue and safranin, respectively.

### 2.5. Antimicrobial Efficacy of the Surface-Functionalized WO_3_-AA NPs

The antimicrobial activity of the prepared WO_3_-AA NPs was investigated against three pathogenic strains, including *Staphylococcus aureus* (ATCC 25923), *Pseudomonas aeruginosa* (ATCC 27853), and *Candida albicans* (ATCC 10231) along with plain WO_3_ and separate AAs as controls. Microplate reader assay was applied in this test, where pre-culture was prepared by overnight cultivation of the three organisms on nutrient broth medium at 37 °C. Tissue culture plates (flat-bottom 96 wells) were inoculated separately with 100 µL of the diluted tested organism (106 CFU/mL) and serially diluted compounds (5–80 µg) to a final volume of 200 µL. The microtiter plates were incubated overnight at 37 °C and measured at 600 nm with a microplate reader. Antimicrobial results were expressed as minimum inhibitory concentration (MIC), defined as the lowest concentration that eliminated all cells. 

### 2.6. In Vitro Anticancer Studies

#### 2.6.1. Cell Culture and Media

Human normal amnion (WISH), human cells derived from breast adenocarcinoma (MCF-7), human epithelial cells derived from colorectal adenocarcinoma (Caco-2), and human hepatocyte carcinoma (HepG-2) cell lines were obtained from VACSERA (Cairo, Egypt). DMEM and RPMI-1640 media were obtained from Lonza (Lmd. Co., Basel, Switzerland). Human WISH and Caco-2 cell lines were cultured in DMEM media supplemented with 10% FBS (Fetal bovine serum). However, MCF-7 and HepG-2 cell lines were cultured in RPMI-1640 media supplemented with 10% FBS.

#### 2.6.2. Cytotoxicity Evaluation Using MTT Assay

The anticancer effect of the functionalized WO_3_-AAs NPs was evaluated through MTT (3-[4,5-Dimethylthiazol]-2,5-Diphenyltetrazolium bromide) assay protocol using normal and cancerous cells as described previously by El-Fakharany et al. [48]. Briefly, both normal and cancer cells (1.0 × 10^4^/well) were seeded in four sterile 96-well microplates and incubated for 24 h with complete culture media. Serially diluted functionalized WO_3_-AAs NPs at concentrations of 0.1–0.5 mg/mL were added to the cultured cells in triplicates and incubated for 48 h in a 5% CO_2_ incubator. Then, cells were washed 3 times with fresh media, and 200 μL of MTT solution (0.5 mg/mL) was added to each well for 3–5 h. After the incubation period, MTT solution was substituted with 200 μL of DMSO for dissolving formazan crystals. The optical density was measured at 590 nm using a microplate reader. The IC_50_ value (half maximal inhibitory concentration) of each WO_3_-AAs NPs that cause killing of 50% cells was determined by the Graphpad Instate software 6.0 and values of selectivity index (SI) which defined as the ratio of IC_50_ of normal cells versus cancer cells were also estimated as described previously by Abu-Serie and El-Fakharany [49].

#### 2.6.3. Nuclear Staining

The anticancer and apoptotic effect of the WO_3_-Cys and WO_3_-Trp NPs against HepG-2 cells was studied by fluorescent nuclear stain method using propidium iodide (PI) dye in comparison with untreated reference cells. HepG-2 cells were incubated in triplicate and treated with different NPs at different concentrations of 0.1, 0.2, and 0.3 mg/mL, as described above. Both untreated and treated cells were washed 3 times with fresh media for removing debris and dead cells and then fixed with 4% paraformaldehyde for 10 min at room temperature. After the fixing step, permeabilization of the cells with 0.5% Triton X-100 and 3% paraformaldehyde was performed for 1 min. The PI dye at a concentration of 10 μg/mL was added to stain the cells for 20 min. The cells were examined and imaged by a fluorescence phase-contrast microscope (Olympus, Tokyo, Japan) using an excitation filter of 480/30 nm [50].

#### 2.6.4. Effect of the Surface-Functionalized WO_3_-AA NPs on Gene Expression

The effect of the prepared WO_3_ and surface-functionalized WO_3_-AA NPs on the expression level of some tumor genes, including transcription factor 2 gene (E2F2), tumor suppressor gene (p53), and oncogene (Bcl-2) was determined in human MCF-7, Caco-2, and HepG-2 cells using qPCR method according to the manufacturer’s instructions for the SYBR green kit (BiotechCo., Ltd., Carrollton, TX, USA). After treatment, the cancer cells with the modified WO_3_-Cys and WO_3_-Trp NPs at IC_50_ concentrations for each modified NP as described above, total RNAs of each untreated and treated cancer cell line were extracted using the protocol of Gene JET RNA Purification Kit (Thermo Scientific, Waltham, MA, USA). Real-time PCR of each cDNA was carried out by a master mix of SYBR green kit using specific primers (Forward/Reverse) as follows: 5′-GCATCCAGTGGAAGGGTGTG-3′/5′-ACGTTCCGGATGCTCTGCT-3′ for E2F2 gene, 5′-TAACAGTTCCTGCATGGGCGGC-3′/5′-AGGACAGGCACAAACACGCACC-3′ for p53 gene, and 5′-TCCGATCAGGAAGGCTAGAGTT-3′/5′-TCGGTCTCCTAAAAGCAGGC-3′ for Bcl-2 gene. The change of the gene expression level for each cell line before and after treatment was determined by using the equation of 2^−ΔΔCT^(2ˆ(−delta delta of the threshold cycles (*CTs*)).

### 2.7. Docking Studies

Molecular docking studies were carried out for simulating the interaction of WO_3_ NPs and WO_3_-AAs into three important cancer developing proteins, namely, Bcl-2, p53, and E2F2. The crystal structures of the three aforementioned proteins were downloaded from the protein data bank (PDB) website with ID 2W3L, 3ZME, and 5TUU, respectively. Molecular operating environment (MOE) software was used to conduct molecular docking. The downloaded protein was prepared by 3D protonation, deletion of water molecules and unwanted peptide chains, and energy minimization. The pocket was obtained by isolating the molecular surface around the binding site (within 4.5 Å near the ligand atom). Validation was carried out by redocking the crystallized ligand. Rigid protocol docking (default protocol) was selected. The docking protocol is considered valid when the root mean square deviation (RMSD) of the docking pose compared to the co-crystal ligand position is about 1.5 Å or less. To prepare WO_3_ NPs and modified WO_3_-AAs for docking, the structure of the studied compound was built on MOE. 3D protonation was selected, then energy minimization (force field: MMFF94x) was applied. The prepared compounds were added to the created database, which was then selected for docking according to the default protocol in which the bond rotation method generated ligand conformations. The conformers are placed on the site with the Triangle matcher method [51].

### 2.8. Statistical Analysis

All data were expressed as mean ± SE (standard error of the mean) of three different experiment measurements. The statistical significance was assessed by the multiple comparisons of Tukey’s post-hoc analysis of variance (ANOVA) using the SPSS16 program (IBM, Armonk, NY, USA). Differences of the results were considered statistically significant at (*p*)-values < 0.05. 

## 3. Results and Discussions

### 3.1. Characterization of WO_3_ NPs and WO_3_-Amino Acid (WO_3_-AAs) Complexes

The WO_3_ NPs were prepared from Na_2_WO_4_ solution eluted through a strong cation exchange resin where the resulted yellow solution was precipitated and dried at 70 °C. The surface of dried NPs was functionalized with Trp or Cys (Figure 1). Moreover, Figure 1 revealed that each building unit of the WO3 nanosheet contains two terminal (W = Ot) groups, which were available to interact with two amino acid units through the amine group. Therefore, we suggested that the molecular ratio between WO_3_ nanosheet and amino acids was 1:2. The appearance of synthesized WO_3_·H_2_O and modified WO_3_-AAs at various magnifications was explored by SEM (Figure 2). The SEM photos presented WO_3_·H_2_O in separated nanosheet morphology with a homogeneous size and dimensions determined to be about 0.14 µm in average thickness and 0.33–0.74 µm length/width using software of the SEM-Smile View program (Figure 2). The functionalized WO_3_-AAs complexes under the SEM microscope presented as a coagulated structure formed from the WO_3_·H_2_O sheets linked with each other by the amino acid to form agglomerated WO_3_·H_2_O nanosheets/AAs complexes (Figure 2). In a related work, individual amino acids, especially for histidine, were proposed for better understanding the NPs and protein interaction [52]. Though several mechanisms have been proposed for the interaction of protein and amino acids on the surface of NPs, the exact mechanism is not fully understood [30]. As per the literature, the corona-like structure resulting from the proteins and NPs interaction is a well-established theory for elucidating the different biological behaviors of NPs in the biological media [52,53]. It is worth mentioning that the WO_3_·H_2_O keeps on its sheet morphology in the resulting WO_3_-AAs complexes.

Energy-dispersive X-ray spectroscopy (EDS) was applied to reveal the elemental analysis of the materials. The analysis demonstrated that the synthesized WO_3_·H_2_O is free from impurities, where powder is composed of %At (W = 42.10 and O = 57.90) as shown in Figure 2. On the other hand, EDS examination of the surface-functionalized WO_3_ (WO_3_-Trp and WO_3_-Cys) revealed the presence of carbon atoms (Figure 2), indicating that the carbons came from the two amino acids.

The FT-IR spectra were used to identify the structural-functional groups of WO_3_·H_2_O and WO_3_-AAs complexes (Figure 3A). The IR spectrum of WO_3_·H_2_O (Figure 3A) revealed a broad band at 3387 cm^−1^ corresponding to the stretching motion of O–H, the band at 1624 cm^−1^ is due to in-plane bending δ(H–O–H) of the water molecule. Moreover, the weak band at 938 cm^−1^ characterized the stretching of W = O_t_ (where O_t_ is the terminal oxygen). The intense sharp band at 666 cm^−1^ revealed the stretching of W–O. Upon reacting with Trp or Cys, the peaks related to the stretching motion of O–H become more intense, broader, and shifted to 3428 and 3442, respectively (Figure 3A). Furthermore, the bending δ(H–O–H) of the water molecule band becomes more intense and shifted to 1650 cm^−1^ with Trp and to 1644 cm^−1^ in the case of Cys. The results (Figure 3A,B) revealed several new bands at 1434, 1272, 1115, 1063, 984, 888, 627, 525 cm^−1^ for WO_3_-Trp complex and 1423, 1272, 1114, 1061, 983, 888, 654, 532 cm^−1^ for WO_3_-Cys complex where the intense sharp band at 666 cm^−1^ disappears (Figure 3). These results asserted the successful interaction between the two amino acids and WO_3_ NPs. 

The thermal stability of WO_3_·H_2_O and WO_3_-AAs complexes was tested using TGA analysis, as seen in Figure 3B. The WO_3_·H_2_O present two main decomposition stages 64–90 °C (2.5%) related to the evaporation of adsorbed water 197–249 °C (4.5%) corresponding to the release of intermolecular water (Figure 3B). Attributed to modification of the WO_3_ NPs with tryptophan (WO_3_-Trp) or cysteine (WO_3_-Cys), the TGA properties of the produced composites were consequently altered from the original WO_3_ (Figure 3B). For both composites, desorption of solvent, physically bonded water, and surface hydroxyl groups were assessed through the loss in the mass in the region below 200 °C. At temperature ≥ 250 °C, the two composites presented smoothly and decreased in their masses. This is related to the degradation of the organic materials on the substrate of WO_3_ NPs (Figure 3B). This decomposition behavior presented suitable thermal stability for the organic modifier, confirming the interaction and linking of the amino acids to the surface of WO_3_ NPs.

### 3.2. Dye Decolorization Activity of Prepared WO_3_ and Modified WO_3_-AAs NPs

The ability of modified WO_3_-Trp and WO_3_-Cys NPs toward synthetic dye remediation was evaluated against some synthetic dye solutions (100 ppm/final conc.) and compared to plain WO_3_ NPs and separate AAs. Among the tested dye solutions, WO_3_-AAs revealed high dye removal capacity against methylene blue and safranin dyes (Figure 4) compared to WO_3_ NPs and separate AAs. The decolorization rate of the two dyes was further studied by following up dye decolorization for 6 h, where samples were withdrawn and measured using a UV-visible spectrophotometer with 1 h interval (Figure 4). The results indicate great enhancement in the dye removal capacity for WO_3_-Trp and WO_3_-Cys modified NPs toward methylene blue and safranin dyes with complete dye removal (100%) after 6 h compared to plain WO_3_ that revealed moderate activity with 47 and 70% of dye decolorization for methylene blue safranin, respectively. In solutions, cationic dyes particles (as methylene blue and safranin) usually aggregate on the large negatively charged surface of the WO_3_ NP. When cationic dyes are available in high concentration, more dye particles are adsorbed to the WO_3_ NP surface, which increases their density and the WO_3_-dyes aggregate perception started [54]. The amino acids (Trp and Cys) on the modified WO_3_ NP may enhance the first step of dye-NP aggregation through their abundant COO^−^ groups that bind strongly to the positively charged dye particles, revealing higher dye removal activity compared to the original WO_3_ NP. The surface modification of WO_3_ NP was reported to enhance its dye removal activity [55]. Dinari and his colleagues reported enhancement in methylene blue dye removal by a photocatalytic modified multi-walled carbon nanotube/WO_3_ compared to WO_3_ NP [56]. The two separate AAs revealed no detectable decolorization activity toward both dye solutions (Figure 4). Though the decolorization rate of WO_3_-Trp for both dyes was faster than WO_3_-Cys, the time required for complete degradation of methylene blue and safranin dyes was the same (6 h).

### 3.3. Antimicrobial Activity of Prepared NPs and Modified Amino Acid-NPs

The widespread microbial pathogens for multi-resistance implies a necessity for efficient broad-spectrum antimicrobial molecules with lower microbial resistance induction capacity. Antimicrobial activity of the WO_3_-AAs NPs was investigated against three pathogenic strains, including *Staphylococcus aureus* (ATCC25923), *Pseudomonas aeruginosa* (ATCC 27853)*,* and *Candida albicans* (ATCC 10231) along with plain WO_3_ and AAs as controls (Table 1). The results (Table 1) showed that pure WO_3_ showed no antimicrobial activity against the applied pathogens, which, consistent with other’s findings [55], reported the inability of pure WO_3_ to inactivate pathogenic bacterial cells even after under visible light illumination for 6 h. Separate Cys and Trp revealed considerable antimicrobial activity compared to that of Trp and WO_3_ NPs. The results also indicate a synergistic enhancement in the antimicrobial activity of WO_3_ NPs via modification with the applied amino acids as WO_3_-Cys revealed higher broad-spectrum antibacterial activity toward *Staphylococcus aureus* and *Pseudomonas aeruginosa* compared to MIC values of Cys(85.82 and 112.7 µg/mL), with limited antifungal activity toward *Candida albicans* (105.9 µg/mL) compared to that of Cys (115 µg/mL). The results, consistent with other’s findings, stated the potency of Cys and Cys-rich peptides as promising antibacterial agents [46,57]. Recently, short peptides and amino acid-rich biomolecules rise as promising bacteriostatic agents that most likely target outer bacterial membrane rather than intracellular bacterial structures [46,58]. The thiol side chain in cysteine is highly reactive and hypothetically initiates the first interaction toward bacterial cell membranes [46,59]. On the other hand, WO_3_-Trp also revealed broad-spectrum antibacterial activity toward *Staphylococcus aureus* (MIC 72.5 µg/mL) and *Pseudomonas aeruginosa* (MIC 104.7 µg/mL) with strong antifungal activity toward *Candida albicans* (42.8 µg/mL). The role of Trp as a hydrophobic AAs in antimicrobial-peptides was asserted [47,60] through initiating the first and strong anchoring to the bacterial membrane and affect the interface region of lipid bilayer leading to cell membrane destruction.

### 3.4. Anticancer Activity of the WO_3_ and Surface Functionalized WO_3_-AAs NPs

MTT assay was applied to determine cell metabolic activity for testing the cytotoxicity of a certain compound or drug. Herein, we evaluated the anticancer effect of the modified WO_3_-Trp and WO_3_-Cys and NPs against various kinds of cancer cells, including MCF-7 caco-2 and HepG-2 cell lines in comparison with free forms of Cys and Trp AAs along with original WO_3_ NPs. We also investigated the effect of these compounds on WISH cells as a normal cell line. Our results indicate that the highest values of IC_50_ were found to be for normal cells, which refer to the highest safety of the applied modified NPs. As presented in Table 2, IC_50_ values for the modified WO_3_-Cys and WO_3_-Trp NPs against WISH cells were near those free forms of Cys, Trp, and WO_3_ NPs, which highlight their safety. 

Furthermore, MCF-7 cells are the most sensitive cells to treatment with both modified NPs. The modified WO_3_-Cys NPs had an anticancer activity against MCF-7, Caco-2, and HepG-2 cells after treatment for 48 h at IC_50_ values evaluated to be 92.53 ± 5.2, 162.9 ± 7.5, and 184.5 ± 12.4 μg/mL, respectively, with SI values of 22.01 ± 0.01, 12.49 ± 0.02, and 11.04 ± 0.01, respectively. While after treatment with WO_3_-AAs NPs for 48 h, values of IC_50_ were calculated to be 147.5 ± 9.7, 176.3 ± 10.1, and 202.6 ± 15.3 μg/mL against the same cell lines, respectively, with SI values of 14.98 ± 0.03, 12.54 ± 0.02, and 10.91 ± 0.17, respectively. Collectively, the results (Table 2) indicate that WO_3_-Cys NPs displayed an anticancer activity higher than WO_3_-Trp NPs against all tested cells. The interaction of bare NPs to cellular components is indirect [32], where l-cysteine is known to improve the cellular uptake [61], hence the high activity of the WO_3_-Cys NPs may be attributed to the high protonation effect of its SH group toward many intracellular proteins [62]. The results are in agreement with Kato et al. [63], which demonstrated the role of l-cysteine in suppressing cancer development through regulating the excess activity of kinases that enhance cancer progression. In addition, there was a significant increase in the anticancer activity of the modified WO_3_-Cys and WO_3_-Trp NPs against all tested cancer cells in a dose-dependent manner with high selectivity to cancer cells and high safety against normal cells (Figure 5). Figure 5 also confirms the apoptotic effect of the modified WO_3_-AAs NPs and reveals the proportional changes in HepG-2 morphology using the nuclear stain method. HepG-2 cells were observed to lose their normal shape after treatment with both modified WO_3_ NPs for 48 h as compared to reference cells. Figure 6 also indicates more fragmented HepG-2 chromatin with condensed nuclei in a dose-dependent manner. Furthermore, observed chromatin fragmentation, shedding of apoptotic bodies, and nuclear condensation were considered the major properties of the apoptotic pathway, which became more noticeable with increasing the dose of treatment than untreated cells. Figure 7 also confirms the effect of the modified WO_3_-Cys and WO_3_-Trp NPs on expression levels of E2F2, p53, and Bcl-2 genes in MCF-7, Caco-2, and HepG-2 cells as evaluated by real-time PCR in comparison with untreated cells as control and 5-fluorouracil (5-FU) treated cells as a standard drug. Figure 7 indicates that levels of expression of E2F2, Bcl-2 genes were found to be suppressed and down-regulated after treatment with both modified NPs more than 5-FU-treated cells. Furthermore, the expression level of the Bcl-2 gene was reduced in the treated cancer cells by around 2 folds more than control cells in contrast to the p53 expression level that enhanced over 5–8 folds in treated cells compared to untreated controls. Collectively, Figure 1, Figure 2 and Figure 3 indicate the superiority of modified WO_3_-Cys NPs toward all tested cells over that of the modified WO_3_-Trp NPs.

### 3.5. Molecular Docking Studies

Molecular docking studies were carried out for the three compounds into three proteins, namely Bcl-2, p53, and E2F2, which are of crucial roles in cancer development [64,65,66,67]. The purpose was to get insights into the proposed targets of the studied compounds and to explore to what extent the modified NPs can accommodate the pocket and form interactions with the essential amino acid residues. The crystal structures of the three aforementioned proteins were downloaded from the protein data bank (PDB) website with ID 2W3L, 3ZME, and 5TUU, respectively. 

For the Bcl-2 gene, the obtained results showed the significance of the binding of WO_3_-Trp to Bcl-2 through an arene proton interaction with the essential amino acid residue Tyr67 in the same manner as the ligand (Table 3). Further, an arene proton interaction was observed with the side chain of Met74. Additionally, it showed two hydrogen bonds (2.08 and 2.21 Å) with guanidine moiety of Arg105 (Figure 8A). At the same time, WO_3_-Cys NPs showed three hydrogen bonds with guanidine moiety of Arg88 (distances, 2.15, 2.16 and 2.40 Å). Moreover, an electrostatic interaction was observed between the thiol group of cysteine and alpha proton of Met74 (Figure 8B). While WO_3_ NPs showed four hydrogen bonds with Bcl-2 protein, two of which were with the side-chain carboxylate of Glu95 (1.97 and 2.13 Å) and the other two were of distances of 2.21 and 3.23 Å with backbone oxygen and mercapto group of Met74, respectively. Referring to p53 protein, the ligand was found to interact with three residues, Asp228, Cys229, and Thr230. The most significant binding pattern with p53 was that of WO_3_-Cys NPs, which accommodated the pocket showing interactions with the essential residues Cys229 and Thr230 as shown in Figure 8C. Electrostatic interaction with the beta carbon of Cys229 was observed. Two hydrogen bonds of 1.83 and 2.50 Å length were formed with side chain OH and backbone NH of Thr230, respectively. Additionally, it showed an electrostatic interaction with the alkyl sidechain of Leu145. The second most promising interaction with p53 was that of WO_3_ NPs showing a hydrogen bond (distance: 2.26 Å) with the sidechain OH of the essential residue Thr230. Moreover, electrostatic interaction with the backbone oxygen of the effective residue Asp228 was observed. Further hydrogen bond (distance: 2.46 Å) was formed with the backbone oxygen of Val147 (Figure 8D). On the other side, the modified WO_3_-Trp was unable to accommodate the pocket of p53, showing a different binding mode rather than the ligand. However, it formed one hydrogen bond with Thr150 and another with Pro151. It also showed arene proton interactions with Thr150, Pro222, and Pro223.

Finally, the E2F2 protein was reported to have four domains: DNA-binding domain, a dimerization domain, a transactivation domain, and a pocket protein binding domain [68]. The amino acid residues from 86 to 195 construct the dimerization domain. These residues are responsible for the formation of heterodimers with a DP family protein [68]. 5TUU at the PDB website is the crystal structure of the E2F2-DP1 coiled-coil. Docking results into this protein showed three hydrogen bonds for WO_3_-Trp NPs as illustrated in Figure 8E. The first (distance: 2.65 Å) was with guanidine moiety of Arg158. The second was of 2.21 Å length, with the backbone oxygen of Ser135. The third one (distance: 2.28 Å) was observed with the backbone oxygen of Thr129. Moreover, an arene proton interaction was formed with Ala138. Regarding the WO_3_-Cys composite, Figure 8F shows four hydrogen bonds, two of which were of 2.07 and 2.17 Å lengths with guanidine moiety of Arg158. Another hydrogen bond (2.19 Å length) can be observed with the backbone oxygen of Ala159. The fourth one was of 3.14 Å length, with the backbone NH of Ala159. WO_3_ NPs showed one hydrogen bond (2.01 Å length) with the sidechain OH of Thr163. It also showed another hydrogen bond of 2.01 Å length with the backbone oxygen of Ala159.

## 4. Conclusions

Tungsten oxide (WO_3_) represents one of the most interesting metal oxides with various potential industrial applications. In this work, WO_3_ nanosheets were prepared in one step through a cation exchange column and characterized via different methods. The surface modification of the prepared WO_3_ NPs with either l-tryptophan or l-cysteine greatly enhanced their biological activities. The prepared WO_3_-Trp and WO_3_-Cys exhibited strong dye removal activity toward methylene blue and safranin dyes with complete dye removal (100%) after 6 h. The antimicrobial activity assessment revealed broad-spectrum antimicrobial activity toward the three applied organisms: *Staphylococcus aureus* and *Pseudomonas aeruginosa,* and *Candida albicans* with higher antibacterial activity from WO_3_-Cys and potent antifungal activity through WO_3_-Trp. The dye removal activity in addition to the antibacterial activity of the newly prepared WO_3_-AA NPs could privilege their application in the dentistry field for dye removal and management of the oral bacterial infection. The results also indicate a remarkable antitumor activity in a dose-dependent manner through WO_3_-Trp and WO_3_-Cys toward all applied tumor cell lines (MCF-7, Caco-2, and HepG-2) attributed to induction of apoptotic effects. The highest antitumor activity was against the breast cancer tumor cell MCF-7. WO_3_-Cys revealed the superior antitumor activity, which could be attributed to the remarkable activity of the thiol group, as indicated in the molecular docking studies. Collectively, the surface modification of WO_3_ NPs with amino acids, especially that of l-tryptophan or l-cysteine, significantly enhanced their biological activities and may expand their industrial applications.

## Figures and Tables

**Figure 1 pharmaceutics-13-01595-f001:**
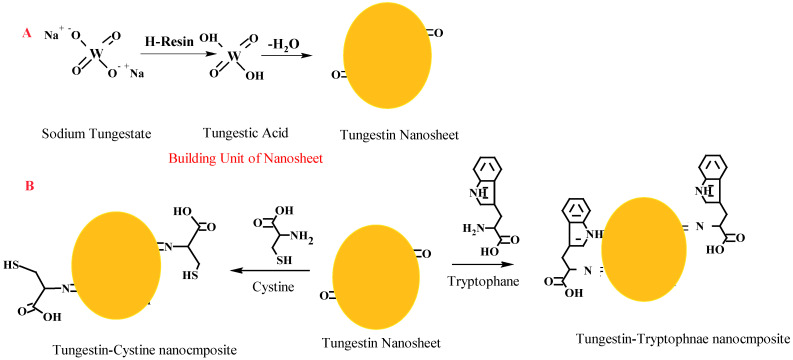
Preparation of WO_3_ NPs through cation resin (**A**) and interaction of the prepared WO_3_ NPs with amino acids l-tryptophan or l-cysteine (**B**).

**Figure 2 pharmaceutics-13-01595-f002:**
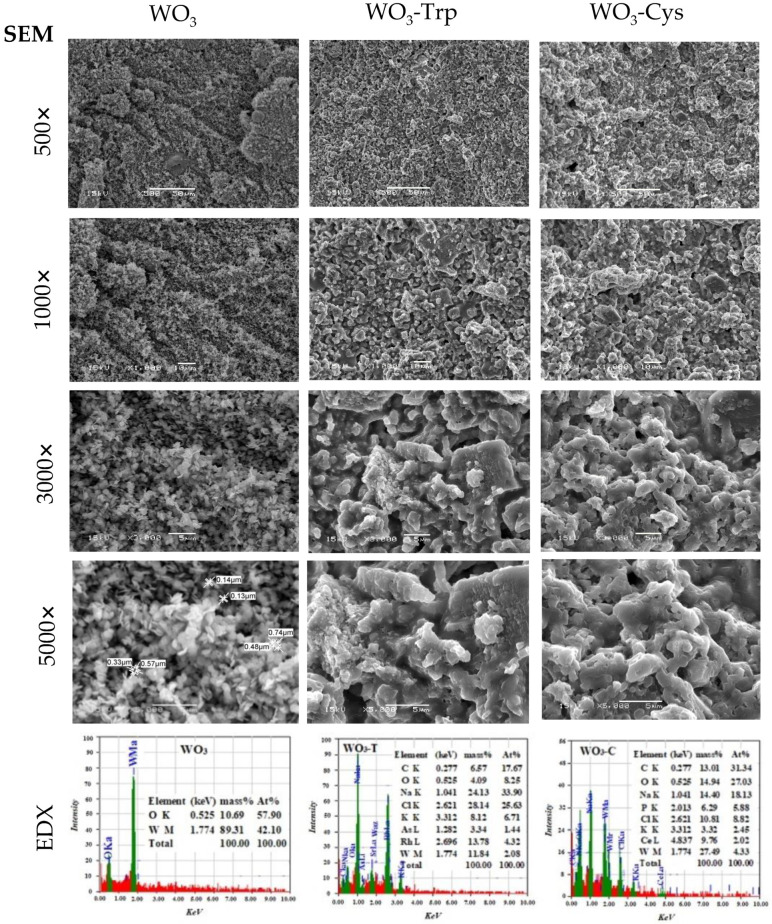
SEM and EDX analysis for prepared WO_3_ NPs (column 1), WO_3_-Trp (column 2) and WO_3_-Cys (column 3) at different magnifications (Element: W, O, C, Na, Cl, K, As, Rh, P, Ce; Shell: K, L, M; Mass%: Mass percent of the element in the materials; At%: Atomic percent of the element in the materials).

**Figure 3 pharmaceutics-13-01595-f003:**
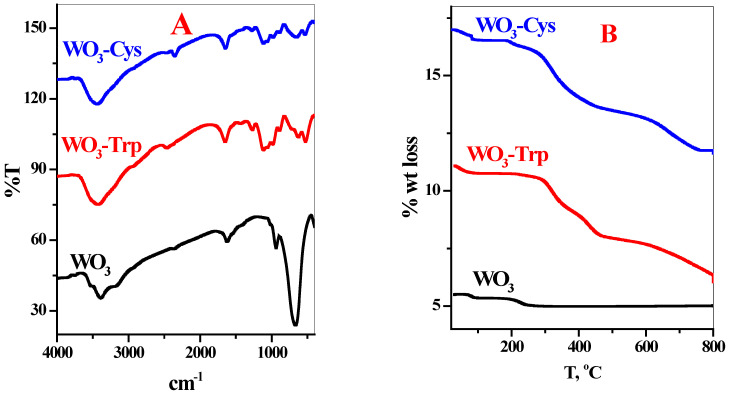
FT-IR spectroscopy (**A**) and TGA analysis (**B**) for the prepared WO_3_ NPs with WO_3_-Trp and WO_3_-Cys modified NPs.

**Figure 4 pharmaceutics-13-01595-f004:**
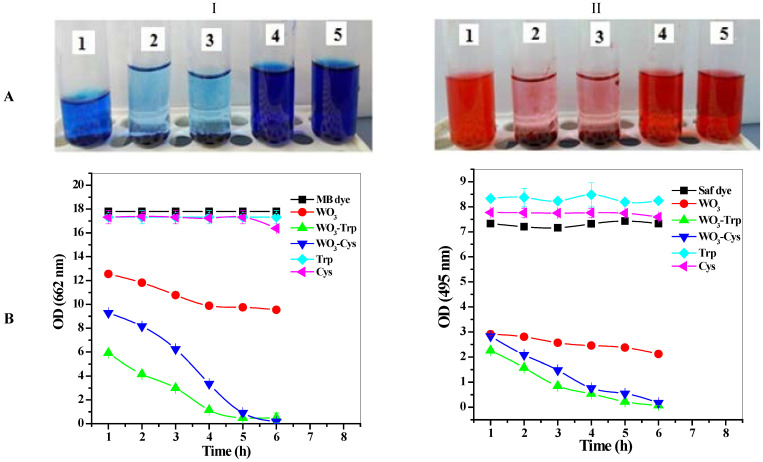
Photographs (**A**) and decolorization percentages (**B**) of dye removal rate for methylene blue (I) and safranin (II) by the prepared modified WO_3_ NPs with Trp and Cys. Methylene blue and safranin at a concentration of 100 mg/mL were decolorized by WO_3_ NPs (1), the modified WO_3_-Trp (2), WO_3_-Cys (3), Trp (4), and Cys (5) for 6 h. The data are presented as mean ± SE and represent the average values from three experiments (*n* = 3).

**Figure 5 pharmaceutics-13-01595-f005:**
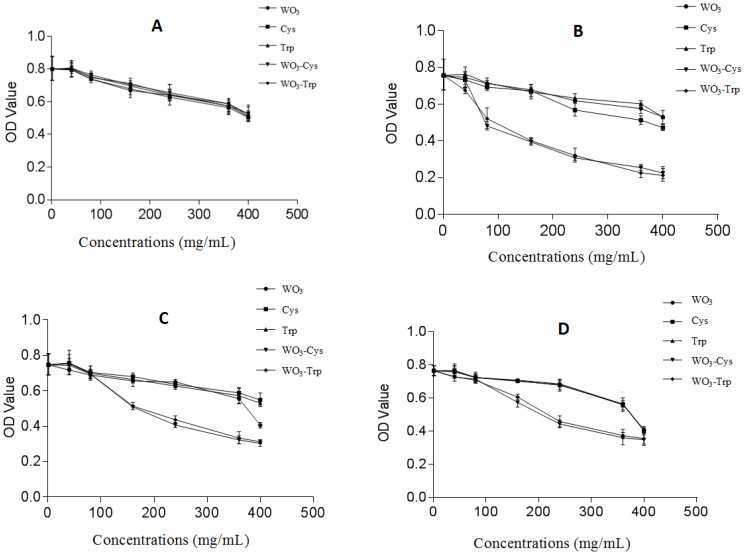
The dependence inhibition effect of the modified WO_3_ NPs on the cell viability of the normal WISH cells (**A**), breast cancer MCF-7 cells (**B**), colon cancer Caco-2 cells (**C**), and hepatoma HepG-2 cells (**D**). Both normal cells and cancer cell lines were incubated with the WO_3_-AAs NPs at different concentrations (0–0.5 mg/mL) for 48 h. All values are expressed as mean ± SE and represent the average values from three experiments (*n* = 3).

**Figure 6 pharmaceutics-13-01595-f006:**
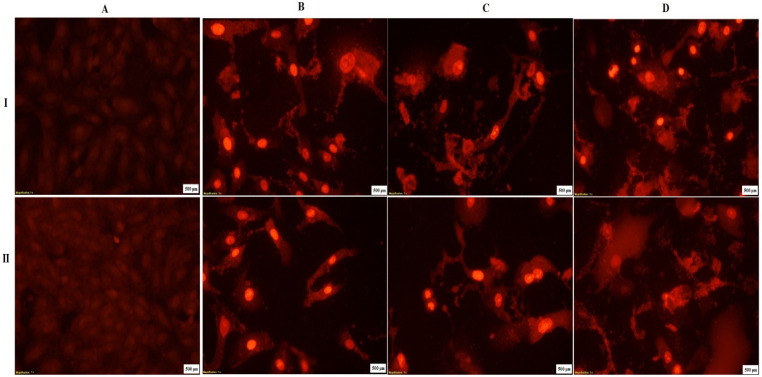
Fluorescence photograph of HepG-2 cells stained with PI dye under a fluorescence microscope after treatment with the modified WO_3_-Cys and WO_3_-Trp NPs (I and II, respectively). HepG-2 cells were exposed to different concentrations of 0.0, 0.1, 0.2, and 0.3 mg/mL (**A**–**D**). The scale bar in all images is 500 μm.

**Figure 7 pharmaceutics-13-01595-f007:**
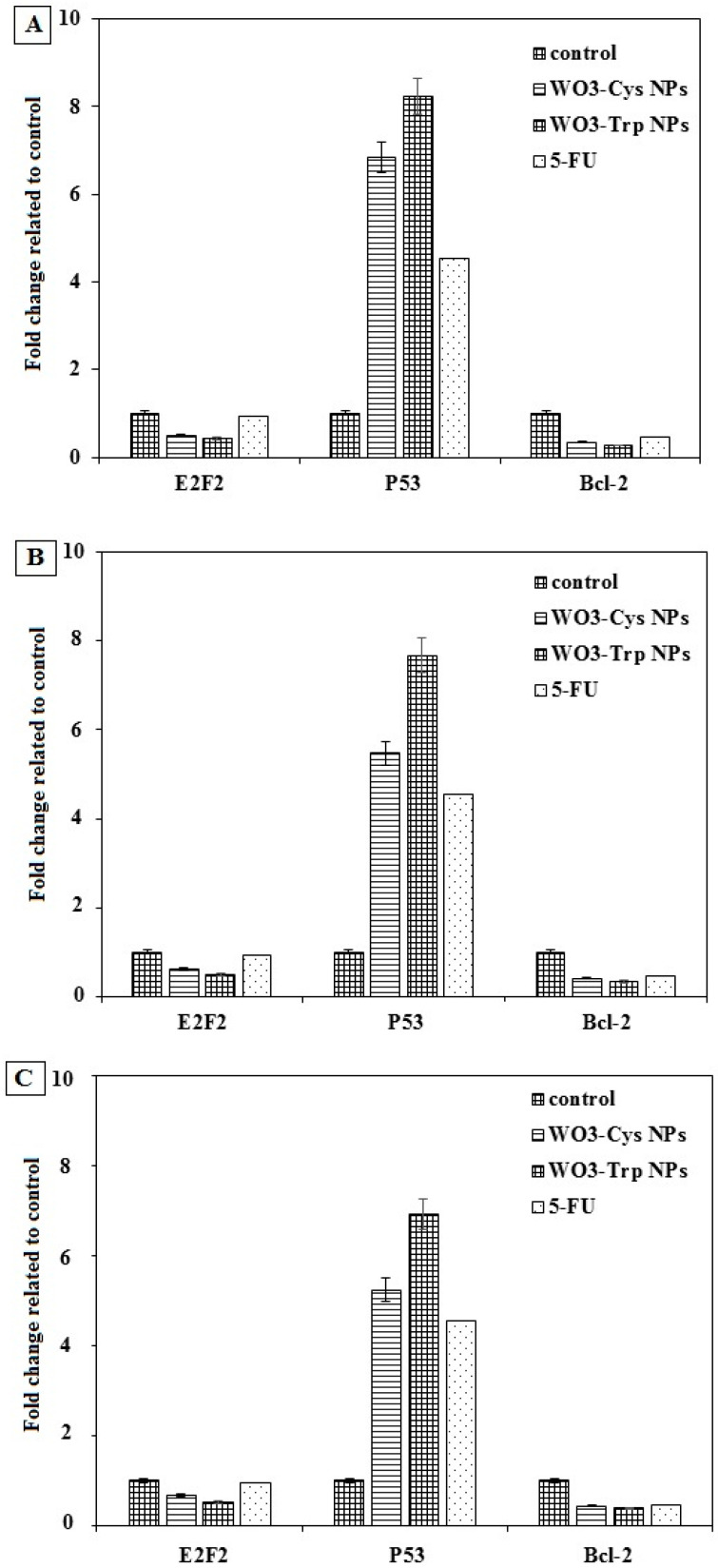
Effects of the modified WO_3_-Cys and WO_3_-Trp NPs on the levels of E2F2, p53, and Bcl-2 mRNA expression in MCF-7 cells (**A**), Caco-2 cells (**B**), and HepG-2 cells (**C**) treated with IC_50_ concentration of each modified NPs and 5-FU for 48 h. The data are presented as mean ± SE and represent the average values from three experiments (*n* = 3).

**Figure 8 pharmaceutics-13-01595-f008:**
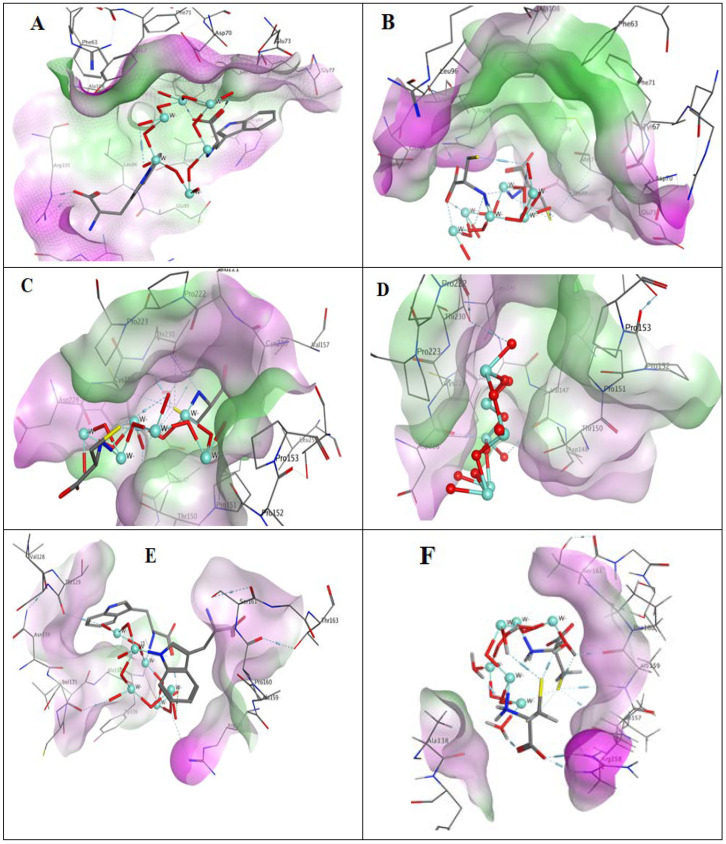
3D interaction of (**A**) WO_3_-Trp nanocomposite with Bcl-2 protein, (**B**) WO_3_-Cys nanocomposite with Bcl-2 protein, (**C**) WO_3_-Trp nanocomposite with p53 protein, (**D**) WO_3_ NPs with p53 protein, (**E**) WO_3_-Trp nanocomposite with E2F2 protein, (**F**) WO_3_-Cys nanocomposite with E2F2 protein.

**Table 1 pharmaceutics-13-01595-t001:** The antimicrobial activity (MIC values) of the modified WO_3_-Cys and WO_3_-Trp NPs against three pathogenic microorganisms in comparison with WO_3_ NPs and free amino acids (Cys and Trp).

Test Organism	WO_3_	WO_3_-Trp	WO_3_-Cys	Trp	Cys
*Staphylococcus aureus*	452.90	72.53	53.14	454.00	85.82
*Pseudomonas aeruginosa*	307.70	104.70	60.52	199.50	112.70
*Candida albicans*	324.90	42.81	105.90	199.50	115.00

The data present the average values from three experiments.

**Table 2 pharmaceutics-13-01595-t002:** EC_100_, IC_50_ (μg/mL), and SI values of the modified WO_3_-Cys and WO_3_-Trp NPs against WISH, MCF-7, Caco-2, and HepG-2 cell lines after treatment for 48 h in comparison with WO_3_ NPs and free amino acids (Cys and Trp).

Sample		WISH	MCF-7	Caco-2	HepG-2
WO_3_	IC_50_	2315 ± 224.3	2805 ± 196.6	2238 ± 210.4	2517 ± 232.5
	SI	-	0.83 ± 0.08	1.03 ± 0.09	0.92 ± 0.1
Cys	IC_50_	2221 ± 159.8	2365 ± 159.7	2635 ± 118.9	2936 ± 102.1
	SI	-	0.94 ± 0.07	0.84 ± 0.05	0.76 ± 0.04
Trp	IC_50_	2064 ± 197.5	2210 ± 156.9	2084 ± 188.4	2958 ± 134.7
	SI	-	0.93 ± 0.07	0.99 ± 0.09	0.69 ± 0.06
WO_3_-Cys	IC_50_	2036 ± 193.2	92.53 ± 5.2	162.9 ± 7.5	184.5 ± 12.4
	SI	-	22.01 ± 0.01	12.49 ± 0.02	11.04 ± 0.01
WO_3_-Trp	IC_50_	2210 ± 166.8	147.5 ± 9.7	176.3 ± 10.1	202.6 ± 15.3
	SI	-	14.98 ± 0.03	12.54 ± 0.02	10.91 ± 0.17

All values were expressed as mean ± SE.

**Table 3 pharmaceutics-13-01595-t003:** Binding free energies of the docked molecules.

Compound.	Binding Energy (Kcal/mol)		
	Bcl-2 (2W3L)	p53 (3ZME)	E2F2 (5TUU)
Ligand	−5.23	−7.14	No ligand
WO_3_ NPs	−6.05	−46.45	−4.03
WO_3_-Trp nanocomposite	−6.54	−38.97	−6.22
WO_3_-Cys nanocomposite	−7.21	−49.00	−3.09

## Data Availability

The data presented in this study are available on request from the corresponding author.

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
