# Peer review of "Novel Nanocombinations of l-Tryptophan and l-Cysteine: Preparation, Characterization, and Their Applications for Antimicrobial and Anticancer Activities"

_pharmaceutics, 2021, doi:10.3390/pharmaceutics13101595_

Round 1

Reviewer 1 Report

In this study, the authors tried to prepare tungsten oxide WO3 nanoparticles (NPs) in a form of nanosheets through acid precipitation using a cation exchange column. The resulting WO3 nanosheets surface was decorated with one of the two amino acids L-tryptophan (Trp) or L-cysteine (Cys) for their dye removal, antimicrobial, and antitumor activities. This study is interesting, however, the insufficient description and poor graphs resolution make it hard to review. The following issues should be addressed prior to its further submission.

The dimension of the nanosheet should be mentioned.

L71, it consider?

L79, AAs’?

L83-86, without verb in sentence

L93, where is scheme 1?

Sec2.1 the elution buffer and flow rate for the packed column?

L100, sec2.2 NP-AA pellets without washing.

L101, where is scheme 1B?

L127, completely eliminated all cells? Is it true? No CFU at all?

Sec2.6.1. cell lines were obtained from VACSERA (Cairo, Egypt), not (VACSERA Cairo, Egypt). Revise all in the text.

L150, Graphpad Instate software 6.0, where to obtain?

L169, qPCR (add ref.)

L173, what is master mix ?

L173, SYBR green kit, where to obtain?

L178, using equation of 2-**TT, what is the equation? It should be listed.

L184, where to obtain MOE software?

L190, RMSD?

L196, Where to obtain Triangle Matcher method? Describe this method in brief.

L215, crona-like?

L217, it importance to note?

Fig.1 too bad of resolution.

Fig.1 How to know the reaction mechanisms of binding these AAs to WO3?

Fig.2, SEM must show the single WO3 sheet dimension.

Fig.2, EDS, the remarks like O K, W M ? mass% and At%?

The WO3 with AAs has a very low WO3 molecular ratio, does this mean whole WO3 is covered by AAs?

Fig3. (a) No differences in these three figures? The peak under 1000cm-1 should be addressed.

Fig.3 (b), why the beginning % wt losses of 3 materials are quite different? e.g., diWO3 begin from 5%?

Fig.4 (b) too bad of resolution.

Fig.4(a) the decolorization of WO3 seem to adsorb the dyes and to precipitate, explain why?

Figs.5-6 too bad of resolution. Fig.6 all the AAs in graph should be clearly listed.

Fig.5 A, the decrease OD value means the same concentration of WO3-AAs are also toxic to normal cell line WISH?

Fig.8 how to know the number of AAs attached to each WO3? The graph seems to draw WO3 in sphere shape, should it be paper sheet in this graph?

Author Response

In this study, the authors tried to prepare tungsten oxide WO3 nanoparticles (NPs) in a form of nanosheets through acid precipitation using a cation exchange column. The resulting WO3 nanosheets surface was decorated with one of the two amino acids L-tryptophan (Trp) or L-cysteine (Cys) for their dye removal, antimicrobial, and antitumor activities. This study is interesting, however, the insufficient description and poor graphs resolution make it hard to review. The following issues should be addressed prior to its further submission.

The dimension of the nanosheet should be mentioned.

We added the dimension of the WO3 nanosheet.

L71, it consider?

This error was corrected.

L79, AAs’?

This error was corrected.

L83-86, without verb in sentence

This error was corrected.

L93, where is scheme 1?

We revised this error.

We changed it to become figure 1A instead of scheme 1.

Sec2.1 the elution buffer and flow rate for the packed column?

The Na2WO3 freely flow through the column using phosphate buffer saline at flow rate of 10 ml/min.

L100, sec2.2 NP-AA pellets without washing.

The pellets were washed several times with double distilled water and phosphate buffer saline.

L101, where is scheme 1B?

Scheme 1B was changed to Fig. 1B.

L127, completely eliminated all cells? Is it true? No CFU at all?

Yes, the sentence is true especially when referred to a defined period of time as mentioned in many references (DOI: 10.1046/j.1469-0691.2000.00149.x) and (DOI: 10.3390/pathogens10020165)

Sec2.6.1. cell lines were obtained from VACSERA (Cairo, Egypt), not (VACSERA Cairo, Egypt). Revise all in the text.

Thank you for your comment.

This error was revised.

L150, Graphpad Instate software 6.0, where to obtain?

It is a free trial.

L169, qPCR (add ref.)

The qPCR was performed according to the manufacturer’s instructions of SYBR green kit

L173, what is master mix?

Master mix is PCR mix containing Taq polymerase, dNTPs, MgCl2 and buffer.

L173, SYBR green kit, where to obtain?

(Biotech Co., Ltd)

L178, using equation of 2-**TT, what is the equation? It should be listed.

The equation was added.

L184, where to obtain MOE software?

It can be requested from chemical computing group at (https://www.chemcomp.com/).

L190, RMSD?

It is root mean square deviation. We added it in the text immediately before the abbreviation.

L196, Where to obtain Triangle Matcher method? Describe this method in brief.

This method is used by MOE software as huge number of conformers were created for each molecule as a result of free rotation of single bonds. Hence this requires a suitable method to place these conformers in the site of binding in a highly systematic manner. According to MOE tutorials, Triangle Matcher method is selected to run this part of docking process.

L215, crona-like?

We corrected this error.

L217, it importance to note?

We corrected this error.

Fig.1 too bad of resolution.

We improved the resolution of this figure.

How to know the reaction mechanisms of binding these AAs to WO3?

From the characterization of the nanomaterial before and after the combination process.

Fig.2, SEM must show the single WO3 sheet dimension.

We improved the resolution.

Fig.2, EDS, the remarks like O K, W M ? mass% and At%?.

We revised the caption of figure 2.

The WO3 with AAs has a very low WO3 molecular ratio, does this mean whole WO3 is covered by AAs?

You are very right, a great amount of the amino acid bonded with the nanomaterial.

Fig3. (a) No differences in these three figures? The peak under 1000cm-1 should be addressed.

referee to the FTIR spectra in the characterization section (Moreover, the week band at 938 cm-1 characterized the stretching of (W=Ot) (where Ot the terminal oxygen). The intense sharp band at 666 cm-1 revealed the stretching (W-O))

Fig.3 (b), why the beginning % wt losses of 3 materials are quite different? e.g., diWO3 begin from 5%?

WO3 is metal oxide material, only possess wt loss for the surface adsorbed water. After modification with amino acids the mainly weight loss related to the amino acid liked to the surface of nanoparticles. And due to quite high amount of the amino acid combined with the nanoparticles, we observed quite difference in the wt loss of the three materials.  

Fig.4 (b) too bad of resolution.

The resolution improved

Fig.4(a) the decolorization of WO3 seem to adsorb the dyes and to precipitate, explain why?

In solutions, cationic dyes particles (as methylene blue and safranin) usually aggregate on the large negatively charged surface of the WO3 NP. When cationic dyes are available in high concentration more dye particles adsorbed to the WO3 NP surface and increasing their density and the WO3-dyes aggregate perception started (https://doi.org/10.1039/C5RA18601C). The amino acids (Trp and Cys) on the WO3 NP may enhance the first step of dye-NP aggregation through their abundant exposed COO- groups that bind strongly to the positively charged dye particles revealing higher dye removal activity compared to the original WO3 NP. 

This explanation was added to part 3.2 with yellow highlight 

Figs.5-6 too bad of resolution. Fig.6 all the AAs in graph should be clearly listed.

The resolution of figures 5 and 6 were improved.

We investigated the WO3-AAs only, because they have an increased SI than other compounds.

Fig.5 A, the decrease OD value means the same concentration of WO3-AAs are also toxic to normal cell line WISH?

But the selectivity indexes of WO3-AAs are increased than other used WO3 or AAs, that’s mean they are selective to tumor cells than normal cells.

Fig.8 how to know the number of AAs attached to each WO3? The graph seems to draw WO3 in sphere shape, should it be paper sheet in this graph?

To make it easy to notice the number of amino acids attached to each WO3, 2D figures were supplied. With respect to the shape of WO3, you are true the shape of WO3 is not planar, it is actually in the most stable puckered shape. It is well known that one nano is equal to 10 Angstroms. The average of bond length, meanwhile is about 1.5 Å.  According to these data, it is applicable to nanosheet to have puckered compounds. 

Reviewer 2 Report

Present work discuss about the synthesis of surface modified tungsten oxide nanosheets for antibacterial and anticancer therapy. following are some pointers

1) Introduction to why they have chosen to modify surface of tungsten oxide nanosheet with cysteine and tyrosine is not provide. What is the rationale behind the use of this surface modification towards the possible improved activity towards bacteria and cancer cell lines

2) Introduction section line 62 to 86 is extremely generalized and does not provide any information on purpose of developing amino acid modified tungsten oxide nano-sheets. Specifically authors referred to tyrosine or Cysteine rich short peptides and abstract talks about simply amino acid modified tungsten peptide. I am confused if authors have short peptide tagged tungsten sheet or amino acid tagged sheets?

 3) what is the purpose of dye remediation or dye removal, with respect to antibacterial or anticancer therapy? Is it an independent observation? Is there any implication with respect to theranostics or drug release model?

4) Authors have provided standard deviation in the antibacterial and anticancer evaluation for table 1, and table 2 respectively. How many sets were considered for taking SEM (standard error of mean) and kindly provide p values for t-tests for subsets.

Authors have mentioned that MTT assay was done in triplicate kindly mention this in figure legend also clarify the sample set for antibacterial investigation.

Also SEM abbreviation co-insides with SEM (scanning electron microscopy), please use SE (standard error) instead of SEM for standard error of mean.

5) Figure 6: scale bar is not visible. And this is not discussed in the text. I am not sure what is the purpose of this experiment? Also provide untreated cells (-ve control)  for relative comparison.

6)  Figure 7, Authors uses SD in their analysis. Please be consistent with your analysis of using SD or SE while presenting your data. Please provide number of sets used for calculation of SD or SE in your data sets.

7) What was the Gibbs free energy of binding for the docked model? I think docking evaluation of organic-inorganic nanocomposite with protein is really interesting and innovative data. But to my understanding the molecular structure of 2D and 3D stabilized structure of ligand is not known in any standard data base like PubChem or Chemspider. Hence it will be useful for authors to provide the 2D structure along with its 3D conformation of WO3-Tryp and WO3-Cys as supplementary, that was used for docking evaluations.

Can you please clarify about the selection of binding pocket location? Line 186 to 187: molecular surface around ligand was isolated for binding. Was there any reference ligand site that was used for docking here? Can you please share x, y, z grid coordinates?

Author Response

Present work discuss about the synthesis of surface modified tungsten oxide nanosheets for antibacterial and anticancer therapy. following are some pointers

  • Introduction to why they have chosen to modify surface of tungsten oxide nanosheet with cysteine and tyrosine is not provide. What is the rationale behind the use of this surface modification towards the possible improved activity towards bacteria and cancer cell lines

Introduction to why they have chosen to modify surface of tungsten oxide nanosheet with cysteine and tyrosine is not provide.

Actually its already provided in the introduction line 80-87.

What is the rationale behind the use of this surface modification towards the possible improved activity towards bacteria and cancer cell lines?

  • The rationale behind the surface modification of the WO3 with amino acids could be summarized in two points: first decreasing the cytotoxicity of the sole nanoparticles and second increasing the scope of the biological activities with safe biological material expected to have strong biological activities

  • Introduction section line 62 to 86 is extremely generalized and does not provide any information on purpose of developing amino acid modified tungsten oxide nano-sheets. Specifically, authors referred to tyrosine or Cysteine rich short peptides and abstract talks about simply amino acid modified tungsten peptide. I am confused if authors have short peptide tagged tungsten sheet or amino acid tagged sheets?
  • Actually we used simple amino acids for modifying the WO3 nanoparticles surface. The whole idea of the current research based upon the recent applications for short peptide in medical filed with growing question about the real reason behind the reported activity is it for the whole peptide or its amino acid contents, and hence we applied the simple amino acid to explain this point.

 3) what is the purpose of dye remediation or dye removal, with respect to antibacterial or anticancer therapy? Is it an independent observation? Is there any implication with respect to the ranostics or drug release model?

Because we wanted to reveal that these prepared WO3-AAs can be used in many medicinal applications such as anticancer and in dentistry field for dye removal and management the oral bacterial infection.

- the clarification was added to the conclusion part.

4) Authors have provided standard deviation in the antibacterial and anticancer evaluation for table 1, and table 2 respectively. How many sets were considered for taking SEM (standard error of mean) and kindly provide p values for t-tests for subsets.

N = 3

All tests were performed in triplicates.

Authors have mentioned that MTT assay was done in triplicate kindly mention this in figure legend also clarify the sample set for antibacterial investigation.

Thank you for your comment.

We revised these mistakes.

Also SEM abbreviation co-insides with SEM (scanning electron microscopy), please use SE (standard error) instead of SEM for standard error of mean.

We corrected these errors.

5) Figure 6: scale bar is not visible. And this is not discussed in the text. I am not sure what is the purpose of this experiment? Also provide untreated cells (-ve control) for relative comparison.

The resolution of this figure was improved and the scale bar was added.

This experiment was performed to confirm the apoptotic effect of the prepared WO3-AAs compounds.

We included untreated cells as negative control to the figure.

6)  Figure 7, Authors uses SD in their analysis. Please be consistent with your analysis of using SD or SE while presenting your data. Please provide number of sets used for calculation of SD or SE in your data sets.

Thank you for your comments.

We revised this error, we used SE.

7) What was the Gibbs free energy of binding for the docked model? I think docking evaluation of organic-inorganic nanocomposite with protein is really interesting and innovative data. But to my understanding the molecular structure of 2D and 3D stabilized structure of ligand is not known in any standard data base like PubChem or Chemspider. Hence it will be useful for authors to provide the 2D structure along with its 3D conformation of WO3-Tryp and WO3-Cys as supplementary, that was used for docking evaluations.

Thank you for this valuable comment. The binding free energies of the docked molecules were given in the supplementary data. Also, 2D images for WO3-Tryp and WO3-Cys binding patterns were supplied.

Can you please clarify about the selection of binding pocket location? Line 186 to 187: molecular surface around ligand was isolated for binding. Was there any reference ligand site that was used for docking here? Can you please share x, y, z grid coordinates?

Regarding pocket isolation, we followed the procedures involved in the protocol supplied by the software. Docking methodology at MOE tutorials stated that “By default, graphics will display around ligand or dummy atoms. The Within value specifies the clipping proximity (by default 4.5 angstroms)”. Herein, the two proteins 2W3L and 3ZME already have co-crystallized ligands. Hence, each of them was downloaded from PDB website with its co-crystallized ligand. On the other hand, 5TUU is not have a co-crystallized ligand. So that the essential amino acids involved in activity were identified according to reference number 66 in the manuscript.

With regard to 2W3L active site, the calculated x, y and z coordinates were 27.647, 41.747 and 7.459 respectively. With respect to 3ZME active site, these coordinates were 92.828, 92.479 and -43.242 respectively. Meanwhile, 5TUU active site showed x, y and z coordinates equal -12.148, -10.933 and -16.610 respectively. These coordinates represent the centres of the pockets constituted by the active amino acids.

Round 2

Reviewer 1 Report

1."determined to be about 20 nm in average thickness and 100–300 nm in length/width using ImageJ software"?

Please mark the ImageJ scanning NP range in Fig.2. It is strange that the graphs are in micrometer scale not in nanometer scale? So how to obtain the nanoscale dimension?

2. L71, it consider? it considers

3. Fig.6 From the scale bar of this graph (500 micrometer), the reviewer estimate the particle sizes are about 100-200 micrometer, not in nanoscale.

4. Fig.8 how to know the number of AAs attached to each WO3? Please calculate the molecular ratio of AAs to one WO3 sheet?

5. I don't think the author responds to all comments correctly. A major revision is needed especially for the clarification of particle size. Should it be micrometerscale not nanoscale?

Author Response

1."determined to be about 20 nm in average thickness and 100–300 nm in length/width using ImageJ software"?

Thank you for your valuable comment. We are sorry for this a mistake. We substituted the ImageJ program with SEM-SMileView program which is related to the SEM instrument to detect the diameter and thickness of the WO3 sheet. Therefore, we found that, the WO3 has diameter (0.33-0.74 µm) and thickness (0.14 µm)

Please mark the ImageJ scanning NP range in Fig.2. It is strange that the graphs are in micrometer scale not in nanometer scale? So how to obtain the nanoscale dimension?

We marked the image.

  1. L71, it consider? it considers

We corrected this error.

  1. Fig.6 From the scale bar of this graph (500 micrometer), the reviewer estimate the particle sizes are about 100-200 micrometer, not in nanoscale.

Figure 6 is related to the micrograph of cells.

  1. Fig.8 how to know the number of AAs attached to each WO3? Please calculate the molecular ratio of AAs to one WO3 sheet?

The building unit of the WO3 sheet contains two terminal (W=Ot) groups which were available to interact with two amino acid through the amine group. Moreover, the Fig.1 was modified according to this suggestion.

  1. I don't think the author responds to all comments correctly. A major revision is needed especially for the clarification of particle size. Should it be micrometerscale not nanoscale?

Thank you for your comments.

All manuscript was revised.

Reviewer 2 Report

The present manuscript can be accepted for the publication.

Authors have addressed all the queries raised.

Author Response

The present manuscript can be accepted for the publication.

Authors have addressed all the queries raised.

Thank you for your valuable reviewing.

The manuscript was revised.

Round 3

Reviewer 1 Report

I think the MS is improved and can be accepted.